# AVERAGER STUDENT:
# DISTILLATION FROM UNDISTILLABLE TEACHER

**Reyhan Kevser Keser & Behçet Uğur Töreyin**
Signal Processing for Computational Intelligence (SP4CING) Research Group
Informatics Institute, Istanbul Technical University, Türkiye
{`keserr,toreyin`}@itu.edu.tr

## ABSTRACT

Today, some companies release their black-box model as a service for users, where users can see the model's output corresponding to their input. However, these models can be stolen via knowledge distillation by malicious users. Recently, undistillable teacher (Ma et al., 2021) is introduced in order to prevent the knowledge leakage. In this study, with the aim of contributing to solutions for model intellectual property (IP) protection, we propose a novel method which improves the distillation from an undistillable teacher whose goal is make the distillation difficult for students, with the purpose of model protection. The codes are released at `https://github.com/rkevser/AveragerStudent`.

## 1 INTRODUCTION

Protecting model's IP is critical since stealing these models may cause undesired consequences. First, stolen model can be used for commercial purposes against the model's owner company. Moreover, a malicious user can generate adversarial examples using the stolen model. Furthermore, stealing a model may yield disclosure of private data which is used for the training of the model (Kariyappa & Qureshi, 2020). Although knowledge distillation is a beneficial tool for various applications, it may be used for stealing model's IP by malicious users (Ma et al., 2021). For example, one may acquire original model's functionality and model's training data using data-free knowledge distillation approaches (Lopes et al., 2017). The studies with these aims can be categorized as model stealing or model extraction (Tramèr et al., 2016; Chandrasekaran et al., 2020).

## 2 RELATED WORK

With this point of view, a recent study proposed a method called self-undermining knowledge distillation for model protection (Ma et al., 2021). They introduced a concept called undistillable teacher which hardens the distillation for student model while maintaining a similar performance with a normal teacher. After this work, Kundu et al. (2021) proposed an approach to avoid the decreased performance caused by undistillable teachers. They mainly changed the position of distillation for the teacher model in addition to utilizing auxiliary classifiers and self-distillation approach.

## 3 KNOWLEDGE DISTILLATION

Knowledge distillation (KD) represents the process of transferring knowledge between models by matching the outputs of chosen layers of them. The knowledge obtained from the teacher model is called as soft targets, which are usually transferred in combination with labels, namely, hard targets. The loss function for the student model can be formulated as in 1:

$$L_S = \lambda L_{KD}(z^S, z^T) + (1 - \lambda)L_{cls}(z^S, y) \tag{1}$$

where $\lambda$ is the trade-off between distillation loss $L_{KD}$ and classification loss $L_{cls}$, $y$ represents the label, $z^S$ and $z^T$ are the logits of student and teacher networks, respectively. The distillation loss, $L_{KD}$, can be calculated as in 2:

$$L_{KD} = \sum_b \sum_i L(z_i^T, z_i^S) \tag{2}$$

where $b$ means the batches, $L$ is the loss function, $i$ shows the samples in a batch, $z_i^T$ and $z_i^S$ indicate the logits correspond to sample $i$ obtained from teacher and student, respectively.

## 4 PROPOSED METHOD

It is shown in (Kundu et al., 2021) and (Ma et al., 2021) that undistillable teachers have multiple peaks in their softmax response which is transferred to the student models. It is considered that these peaks may be the main factor that misleads the student models. To alleviate the effects of the multi peaks in softmax response of teachers, we propose transferring the mean of features which have the same labels, as the soft labels. For this purpose, we use features from teacher in a batch manner, to obtain the mean values. Without loss of generality, we use logits as the features in the remaining of the section, where it corresponds to the case of logit distillation (Hinton et al., 2015). The loss function used for training of the student is described in 3 as follows:

$$L_S = \sum_b \sum_i L(\widehat{z}_i^T, z_i^S) \tag{3}$$

where $\widehat{z}$ represents the adjusted logits in a batch, where it consists of the mean of logits obtained from samples with the same labels. $\widehat{z}_y$, mean of logits in a batch for a label $y$, is calculated as in 4:

$$\widehat{z}_y = \frac{1}{N} \sum_x z_{x_y} \tag{4}$$

where $x_y$ show the samples with the label $y$, $N$ is the number of samples that have the label $y$, in a batch. $\widehat{z}_y$ represents the mean of logits correspond to $x_y$.

## 5 EXPERIMENTS AND RESULTS

We follow the experimental setup of the prior work (Kundu et al., 2021). To be precise, we use ResNet-18 and ResNet-50 as teacher models on CIFAR-100 dataset. We utilize ResNet-18, MobileNetV2 and ResNet-50 as student models. Furthermore, we also obtain results for the ensemble scenario as in the prior work. The results are presented in Table 1. Results show that our method always outperforms (Kundu et al., 2021) for all scenarios.

Table 1: Results on CIFAR-100 for the undistillable teachers. KD and Skeptical represent (Hinton et al., 2015)'s and (Kundu et al., 2021)'s approach, respectively. Bold values indicate the best results for each scenario. (RN: ResNet, MN: MobileNetV2, Ens: Ensemble)

| Teacher | | Student | | KD | Skeptical | Ours | Skeptical -Ens | Ours -Ens |
|---|---|---|---|---|---|---|---|---|
| Model | Acc. (%) | Model | Acc. (%) | | | | | |
| RN-18 | 77.55 | RN-18 | 77.55 | 75.00 | 77.33 | **77.39** | 76.38 | **79.69** |
| | | MN | 69.24 | 7.13 | 66.62 | **71.80** | 64.26 | **74.92** |
| RN-50 | 78.04 | RN-18 | 77.55 | 72.28 | 77.25 | **77.53** | 75.48 | **79.62** |
| | | RN-50 | 78.04 | 74.14 | 78.65 | **79.18** | 77.61 | **81.60** |
| | | MN | 69.24 | 7.72 | 66.38 | **70.73** | 62.93 | **74.98** |

## 6 CONCLUSION

In this study, we propose a novel approach to improve the distillation from an undistillable teacher. Results indicate that our method outperforms the compared methods. We believe that our approach will contribute to the solutions for model IP protection problem.

ACKNOWLEDGEMENTS

This work has been supported by The Scientific and Technological Research Council of Turkey (TUBITAK) under the grant number 121E378.

URM STATEMENT

The authors acknowledge that at least one key author of this work meets the URM criteria of ICLR 2023 Tiny Papers Track.

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

## A   APPENDIX

In a real case, it is not known whether the teacher is a normal or undistillable model. Hence, we also evaluate our approach on normal teachers, where distillation results are given in Table 2.

Table 2: Results on CIFAR-100 for the normal teachers. KD and Skeptical represent (Hinton et al., 2015)'s and (Kundu et al., 2021)'s approach, respectively. Bold values indicate the best results for each scenario. (RN: ResNet, MN: MobileNetV2, Ens: Ensemble)

| Teacher | | Student | | KD | Skeptical | Ours | Skeptical -Ens | Ours -Ens |
|---|---|---|---|---|---|---|---|---|
| Model | Acc. (%) | Model | Acc. (%) | | | | | |
| RN-18 | 77.55 | RN-18 | 77.55 | 78.96 | 78.79 | **78.98** | 79.68 | **79.95** |
| | | MN | 69.24 | **75.12** | 71.63 | 73.83 | **75.45** | 75.345 |
| RN-50 | 78.04 | RN-18 | 77.55 | 79.21 | 78.51 | **79.39** | 79.86 | **80.205** |
| | | RN-50 | 78.04 | 79.56 | **80.66** | 80.53 | 81.96 | **82.41** |
| | | MN | 69.24 | **75.28** | 71.76 | 73.42 | **76.32** | 75.455 |

Results show that our approach mostly outperforms (Kundu et al., 2021). It yields an improvement on accuracy up to 2.2 % on CIFAR-100, for the case of normal teacher.

