# OpenReview forum: "Averager Student: Distillation from Undistillable Teacher"
_ICLR.cc/2023/TinyPapers — Submitted to Tiny Papers @ ICLR 2023_

### Official Review · Reviewer_YYVg · 2023-03-22

**Confidence:** 4

**Summary Of Contributions:**

Authors propose a novel method for improving the knowledge distillation from undistillable teacher models. They propose transferring mean of features from teacher model.

**Rating:**

Clear, Correct, and Reproducible (CCR): a submission which meets the reviewing criteria

**Strengths And Weaknesses:**

* **Strengths**
    * A simple and clear approach is used.
    * Objective comparison with baselines is presented.
    * The presented method has outperformed baselines with reasonable improvements.
* **Weaknesses**
    * Qualitative visual predictions are not presented. This is understandable given the space constraints.
    * Code is unavailable at this point.


**Suggested Changes:**

I don't have any suggestions, but I would recommend this work for ICLR discussions.

---

### Official Review · Reviewer_UE7M · 2023-04-03

**Confidence:** 1

**Summary Of Contributions:**

The paper talks about novel method to protect models from being stolen by malicious users. The problem, direction as well as experiments have been presented by the author. They achieve results that outperform the current methods.

**Rating:**

High Potential (HP): a submission which meets the reviewing criteria and has potential to make an impact on the field

**Strengths And Weaknesses:**

Strengths:/
1. A novel method is proposed for the problem/
2. The results clearly outperform the current methods.


**Suggested Changes:**

More experiments and ablation studies can present a clear picture.

---

### Comment · Area_Chair_ybi2 · 2023-06-05
**Ready to archive**

This work meets the threshold for archival, contains the URM statement, and is deanonymized.

---

### Meta-Review · Area_Chair_ybi2 · 2023-04-03

**Recommendation:** Invite to present
**Confidence:** 4

**Metareview:**

Thank you for your submission! As the reviewers have pointed out, this work is well written, easy to follow, and proposes a novel yet simple method for improving distillation from an undistillable teacher. This approach appears to outperform the baseline on multiple different model architectures on the CIFAR100 dataset. The reviewers have also noted that additional experiments may have improved the paper, but are not necessary given the space constraints, so the paper seems to be ready for acceptance at its current state.

**Summary:**

This work proposes a logit adjustment scheme to improve distillation from an undistillable teacher, with the aim of protecting model IP. The reviewers have identified the simplicity and efficacy of the proposed approach, but have also suggested that more experiments and releasing the corresponding code could have been helpful.

**Reason For Not Giving A Higher Recommendation:**

- Not fully reproducible (code not provided)

**Reason For Not Giving A Lower Recommendation:**

- Proposed method outperforms baseline methods
- Experiments and paper are clear, correct, and could be reproduced (with effort)

---

### Decision · Program_Chairs · 2023-04-08

Invite to present